# Identification of Cost-Optimal Measures for Energy Renovation of Thermal Envelopes in Different Types of Public School Buildings in the City of Valencia

**María Esther Liébana-Durán [1,\*], Begoña Serrano-Lanzarote [1]** and **Leticia Ortega-Madrigal [2]**

1   Department of Continuum Mechanics and Theory of Structures, School of Architecture, Universidad Politécnica de Valencia, 46022 Valencia, Spain; apserlan@mes.upv.es
2   Instituto Valenciano de la Edificación, 46018 Valencia, Spain; lortega@five.es
\*   Correspondence: eliebanad@gmail.com

**Abstract:** In order to achieve the EU emission reduction goals, it is essential to renovate the building stock, by improving energy efficiency and promoting total decarbonisation. According to the 2018/844/EU Directive, 3% of Public Administration buildings should be renovated every year. So as to identify the measures to be applied in those buildings and obtain the greatest reduction in energy consumption at the lowest cost, the Directive 2010/31/EU proposed a cost-optimisation-based methodology. The implementation of this allowed to carry out studies in detail in actual scenarios for the energy renovation of thermal envelopes of public schools in the city of Valencia. First, primary school buildings were analysed and classified into three representative types. For each type, 21 sets of measures for improving building thermal envelopes were proposed, considering the global cost, in order to learn about the savings obtained, the repayment term for the investment made, the percentage reduction in energy consumption and the level of compliance with regulatory requirements. The result and conclusions will help Public Administration in Valencia to draw up an energy renovation plan for public building schools in the city.

**Keywords:** public school buildings; energy efficiency; optimal cost; energy renovation; public buildings

## 1. Introduction

It is a fact that the European Union is embarked on a path towards the conversion of economy and society with the aim of locating both of them in a more sustainable territory. A strategic framework is determined to promote a thriving, modern, competitive and climate-neutral economy. Among long-term objectives, a reduction of 90% of emissions by 2050 is included, compared to the levels in 1990 [1]. Currently, 36% of the EU's $CO_2$ gas emissions comes from the building stock, and almost 50% of final energy consumption is used for heating and cooling [2]. Therefore, to achieve these goals it is essential to renovate the building stock, by improving the energy efficiency and fostering total decarbonisation.

The current rate of building renovation is between 0.4% and 1.2%. This means that, in order to reach long-term European targets by 2050, it is necessary to double the rate of interventions in existing buildings [3]. Europe is driving a wave of renovation, prioritizing the improvement of the worst energy-efficient buildings, including schools and hospitals [4].

According to the 2018/844/EU Directive, 3% of public administration buildings should be renovated every year. However, the large number of properties, the lack of financing, information and planning are some of the obstacles found.

After checking some interventions in Spanish schools, the aforementioned drawbacks make performances consequently be carried out in two ways: a comprehensive renovation of each building or a phased renovation. The latter allows simultaneous performance in several buildings by improving a specific element, for example, windows, facades,

heating systems, lighting, etc., or by installing renewable energy equipment. Based on the mentioned points, a simultaneous enhancement in several buildings makes it possible to jointly promote performances, to save time in project preparation and processing, as well as favour a further provision of financial assistance, since the projects are promoted by the public sector.

The main issue is that public administrations do not always have enough data or studies on school buildings or potential improvement scenarios, so there are no results that could be obtained in terms of energy saving. This means that, for example, the same renovation measure is implemented in all types of buildings, without knowing that whether in some of them a reduction of the energy demand before renovating heating systems could be necessary. Moreover, another example is those building types in which improvements in facade insulation could be suitable instead of window replacement.

So as to establish those performances with the greatest reduction in energy consumption at the lowest cost, the Directive 2010/31/EU proposed a cost optimisation methodology. Regarding its implementation in schools, the cost-optimal reports in the EU countries during 2018 [5] show that, whereas for the residential building sector some reference buildings have been established in all countries, for those buildings in the tertiary sector, in particular school buildings, not all countries have drawn up reports on them. Moreover, in those studied, there is no building classification, for example, they are grouped as "educational buildings" or "schools," as in the case of Slovakia and Germany, in which only a single building type is studied. Another example can be found in the Czech Republic with a "nursery school" or in the United Kingdom, with a "secondary school." Ferrara et al. [6] made a review on 88 scientific works based on the implementation of optimal-cost based analysis of calculation methods for designing and optimizing nearly zero-energy buildings in Europe. They show that only 4% of the papers studied include school buildings as case study.

The implementation of this methodology in school buildings shows great potential of growth. In addition, it provides local authorities with specific data on energy saving, maintenance costs, interventions, repayment terms, etc.

Furthermore, some studies on energy renovation in school buildings are worth mentioning. Several of them propose renovation measures for thermal envelopes, heating and lighting systems, use of renewable energy sources, etc. For example, Stocker et al. [7] use a calculation method focused on a standard energy demand with life cycle cost methods. Their results show that the optimal performance according to costs represents a value around 50 to 60 kW h/m2p.a regarding heating and cooling energy demand. Likewise, Dalla et al. [8] implemented cost-optimal methodology in some existing school buildings located in the north-east of Italy. They propose 120 sets of measures, including interventions in thermal envelopes, in systems (photovoltaic system and lighting replacement) and replacement of thermal generators (condensing boiler, biomass boiler or electrical heat pump).

Other authors tried to identify measures that enable to reach nearly zero energy building (nZEB) through an analysis from the cost-benefit perspective, as in the case of Lou et al. [9] who look into energy saving and electricity production schemes in a specific school building by using the building energy set eQUEST. The results show that improvement measures such as high-performance in building thermal envelopes, energy-efficient air-conditioning systems and lighting fixtures, as well as building-integrated photovoltaic panels (BIPV), allow to obtain zero energy buildings. Gaitani et al. [10] analysed some school buildings in terms of energy efficiency and cost optimisation, and designed a comprehensive action plan for renovation, a Technical and Financial Toolkit. Likewise, this study is framed within the European project ZEMedS, focused on the renovation of schools in the Mediterranean area to reach nZEB. With the aim of upgrading school buildings and turn them into nZEB, Ferrari et al. [11] focused their research on criteria laid down for the intervention on historical school buildings officially protected by the Italian Cultural Heritage. They assessed an Italian historical school building, and

proved that the nZEB goals could be reached by retrofitting the building itself through measures compatible with the constraint arising from the protection of cultural heritage, and significantly reducing primary energy consumption. Marrone et al. [12] state that a large number of the Italian school building stock has implemented energy retrofitting measures, but the strategies suggested are often taken according to the best and most common practices (considering average energy saving), but not supported by a proper energy research. They evaluated 80 Italian school buildings by using cluster analysis, so as to provide a methodology capable of identifying the best energy retrofitting measures from the cost-benefit viewpoint. Mora et al. [13] state that a large number of Italian schools were built before the entry into force of energy and seismic regulations. Therefore, they simultaneously studied energy retrofitting and seismic upgrading in one school building.

The European project SHERPA (Share knowledge for Energy Renovation in buildings by Public Administrations) [14], is aimed at strengthening the abilities of public administrations at regional and local level to improve energy efficiency in their public buildings' stock, and reduce CO2 emissions. Soto et al. [15] describe the general auditing protocol devised by SHERPA and illustrate by carrying out an audit in one school building. They conclude that in the case of school buildings, in order to reach nZEB, energy efficiency is not always profitable (unless photo-voltaic energy is produced in situ). However, there are other benefits, such as improving comfort and preparing for the climate change.

In order to ease decision-making in future interventions, Jradi [16] identifies the impact of renovation measures on buildings, once enhancements in school buildings are made.

Before proposing different improvement energy measures, some authors establish a classification of school buildings according to different types based on energy factors, year of construction, building geometry, etc., which enable to find renovation solutions for each type. For example, Arambuela et al. [17] suggest a cluster analysis method that supports the definition of representative architectural types, and the identification of a small number of essential parameters, to assess energy consumption for air heating and the production of hot water in 60 schools in Treviso, Italy. Dimoudi et al. [18] also look into the development of school building types over the time within a Greece region, identify the most representative building types and propose seven improvement scenarios. Likewise, in order to classify the public school buildings in Rome, Santoli et al. [19] make use of data on schools, such as composition, (in terms of number, type and size of buildings), energy label of buildings in property of the municipality, which describe quality in terms of energy consumption for building's thermal envelopes and energy consumption, as heat is transferred from several thermal power plants to school buildings. Katafyogiotou et al. [20] and Castro [21] are authors that should be mentioned as an example of classification models. They propose improvement measures in representative school building types in Cyprus and northern Spain.

Through their analyses, Ferrara et al. [6] define two different methods used for the selection of measures in cost-optimal studies. The first one is a manual approach (selecting a defined number of sets of measures and calculating and comparing the global cost values), the other is an automated search (using computer-generated optimisation algorithms). They also establish two methods for energy performance calculation: one simplified (using simplified methods, for example, the quasi-steady state method defined by the UNI EN 13790 standard, and national implementations) and another dynamic (using dynamic simulation tools that allow detailed and precise energy results). According to this classification, this article uses a manual selection method and a simplified performance calculation method.

This article shows the results of adopting the cost-optimal methodology for housing developed by the IVE (Instituto Valenciano de la Edificación, the Valencia Institute of Building), for school building assessment. Specifically, this study applies this methodology to 3 schools within the city of Valencia, looking into energy performance and proposing a series of sets of improvement measures in thermal envelopes. Each school building is representative of a group in the city. As a result, a tool is obtained to identify the type-

energy saving and $CO_2$ emissions through each set of measures, as well as the global cost over 30 years.

On the other hand, Spanish energy saving regulation, the CTE DB HE [22], sets out a number of requirements or demands for retrofitted buildings. This article also highlights these requirements, and shows to what extent they would be complied with each set of measures proposed.

Finally, the tool or system used and the analysis of the results provide a series of indicators on its usefulness as instruments and data for Public Administration to enable decision-making and planning energy renovation in similar school buildings.

## 2. Materials and Methods

### 2.1. Cost Optimisation Methodology

The methodology used in this study is that established for building energy renovation by the Directive 2010/31/EU on energy efficiency in buildings [23], and the Delegated Regulation 244/2012/EU complementing such directive, in particular the cost optimisation methodology [24]. According to Annex 1 of the regulation, this methodology is structured in the following sections:

1. Setting representative buildings.
2. Identification of energy efficiency measures, as well as improvement measures based on renewable energy sources and/or sets of variants of both types of measures applicable to each reference building.
3. Calculation of primary energy demand, resulting from implementing the measures and sets of measures defined for reference buildings.
4. Calculation of global cost as annual net value for each reference building.

Likewise, in the study for the energy analysis of representative buildings and improvement measures, Ce3X v2.3 is used, the Spanish computer software for energy calculation that also verifies the compliance with the CTE DB-HE.

For global cost calculation, a tool developed by the IVE was used, which had been previously applied to different studies on residential buildings.

### 2.1.1. Buildings under Study

In the city of Valencia there are approximately 90 public primary schools. For this research, general data regarding 135 school buildings was obtained, corresponding to 79 schools.

An analysis of construction and architectural features of these buildings has enabled to group them together into six different building types.

The study is focused on three building types (A, B and C). These were built before the entry of the first Spanish Regulation on thermal characteristics of buildings NBE CT 79 into force, so none of them has thermal envelope insulation, nor have they been recently renovated because of their relative age.

The main factors that differentiate these school building types are structure and date of construction. Type A building was built using brick load-bearing walls and metal joints, and types B and C were built using a concrete structure. In type A the roof was made of ceramic tile mounted on wood, whereas the roof in types B and C was built by using a slab flat or pitched under the layers that make up the roof. Type B and C buildings have wooden windows, whereas type C includes metal windows.

In addition to these construction differences, the design and interior spaces vary according to the three types. Type C buildings are elongated or L-shaped buildings, their facade has a historicist character, and were built between 1945 and 1955. Type B buildings were built in the 1960s and follow the designs of the Modern Movement. They have a greater number of floors and are elongated in shape. Type C buildings were built in the 1970s and they are X or XX-shaped (Figure 1).

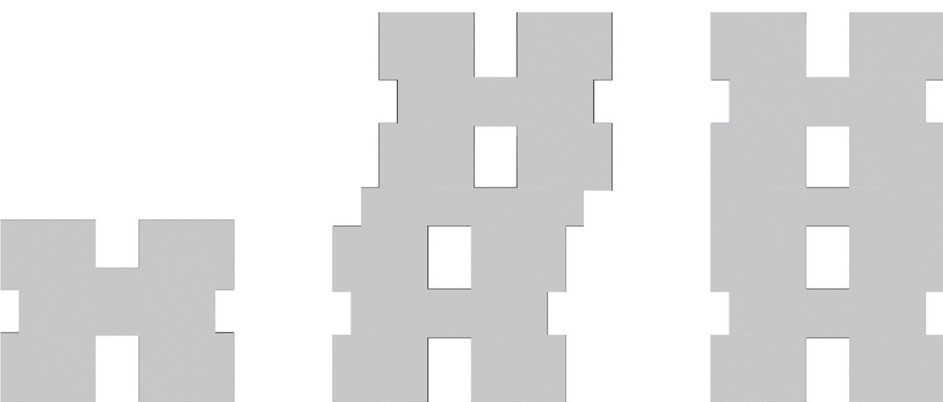

**Figure 1.** Examples of X or XX-shaped buildings.

These three building types (A, B and C), represent a total of 6, 11 and 29 schools respectively, that is to say, 46 school buildings.

For each type, a representative model school building is selected (Figure 2). They were built from school prototypes, adapting the architectural design to the site itself. This makes selected school buildings be representative in the city of Valencia, and also in other towns and municipalities.

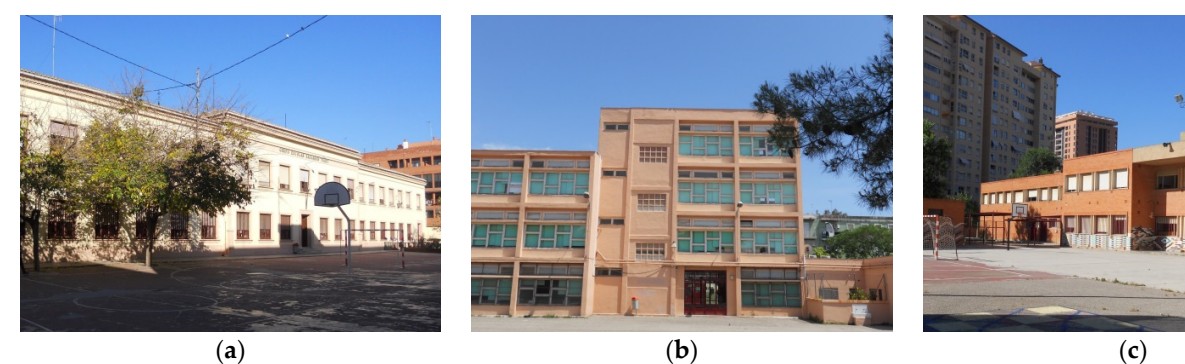

| (**a**) | (**b**) | (**c**) |

**Figure 2.** Representative school buildings in the city of Valencia built before 1979. (**a**) Type A school; (**b**) type B school; (**c**) Type C School.

Type A and B buildings have a high historical value (listed or officially protected), or a unique architectural design. This means that any energy renovation cannot alter the building geometry and exterior design.

Although some of them have undergone minor renovations, such as window replacement, the initial condition of the buildings is taken for the energy study as a reference, and former potential improvements are ignored, so that they serve as a baseline to any building type. Table 1 includes the main characteristics of each building type, and Table 2 shows the thermal transmittance of construction elements in the thermal envelope for each building type in its initial state considered for energy study.

**Table 1.** Specifications and characteristics of building types.

| Thermal Envelope Element | Type A | Type B | Type C |
|---|---|---|---|
| Construction date | 1947 | 1957 | 1975 |
| Constructed area | 1634.77 m$^2$ | 1002.70 m$^2$ | 2595 m$^2$ |
| Living area | 785.89 m$^2$ | 525.42 m$^2$ | 1993.27 m$^2$ |
| Number of floors | 2 | 3/4 | 2 |
| Climate zone | B3/IV | B3/IV | B3/IV |
| Use | 8 h | 8 h | 8 h |
| Roofs | Curved ceramic roof tiles over ceramic brick board, wooden structure, highly ventilated air chamber, false ceiling made of cane, plaster. | Ceramic tiles, mortar, ceramic brick boards, waterproof protection, horizontal air chamber, reinforced with concrete slab floor, plaster. | Ceramic tiles, mortar, waterproof protection, reinforced with concrete slab floor, plaster. |
| Facades | Rendering mortar, brick wall (thickness: 40 cm/50 cm), plaster. | Rendering mortar, double perforated brick wall with a vertical air chamber (11 + 5 + 4 cm), plaster. | Double perforated brick wall with a vertical air chamber (11 + 7 + 7 cm), plaster. |
| Windows and doors | Wooden windows and metal single-glazed doors | Wooden windows and metal single-glazed doors | Metal single-glazed windows and doors. Rolling shutter |
| Heating System | Electric radiator/s in each room | Electric radiator/s in each room | Central heating through diesel oil boiler with multiple water circulation pumps and iron radiator/s in each room |
| Hot Water System | Electric hot water boiler for kitchen and pre-school toilets | Electric hot water boiler for kitchen and pre-school toilets | Electric hot water boiler for kitchen and pre-school toilets |

**Table 2.** Thermal transmittance, U (W/m$^2$K), of construction elements in thermal envelope of representative school buildings in their initial state L0.

| Thermal Envelope Element | Type A | Type B | Type C |
|---|---|---|---|
| Facades | 1.1/0.92 | 2.94/1.33 | 1.41/1.29 |
| Roofs | 4.17 | 2.33/1.79 | 1.79 |
| Windows (single glazed/structure) | 5.7/2.2 | 5.7/2.2 | 5.7/5.7 |

### 2.1.2. Energy Efficiency Measures and Sets of Measures

This study is focused on proposing passive renovation measures, that is to say, measures to implement in the thermal envelope, so renewing air conditioning system, ventilation, lighting, renewable energy, etc., is not considered.

The buildings selected represent a group of school buildings according to specific architectural aspects, but many of them are part of a school complex, where other buildings are introduced, such as classrooms, canteens, gymnasiums, etc. This involves that requirements differ according to different typology, although the design and construction time of the main building are similar.

On the other hand, facility renovations are driven by the time when the existing ones break down or when a necessary renovation due to obsolescence is convenient and affordable. It would be unrealistic to plan the sudden replacement of all the facilities existing in school buildings. For all these reasons, the energy renovation measures proposed are aimed at reducing the energy demand in buildings.

The passive measures outlined are window and door replacement (W), including solar protection elements, and insulation upgrading in facades (F) and roofs (R). These measures

can be implemented separately or combined with each other, resulting in the improvement of seven elements or combinations of elements of the thermal envelope.

In addition, three levels of energy demand are considered in relation to the transmittance of thermal envelope elements (L1, L2 and L3). These values are set according to specific regulatory requirements.

In particular, the intermediate level, L2, includes the minimum values set by current Spanish regulations on energy efficiency, the CTE DB-HE 2019 [22]. Level L3 corresponds to the guideline values of transmittance provided in Annex E of the aforementioned regulation, for pre-dimensioning construction solutions in private residential buildings. With these values, the requirements established for the global heat transmission ratio through the thermal envelope are fully met.

The strategic energy renovation in school buildings introduces a first level, L1, whose transmittance requirements are less restrictive than those set by current regulations. Thus, transmittance values in this level meet minimum requirements set by the same regulations in its initial version CTE DB-HE 2006 [25]. This is proposed with the aim of analysing whether requirements established by the former regulation reach cost-optimal results similar to those obtained with the conditions currently required.

Transmittance for each requirement level is shown in Table 3.

**Table 3.** Thermal transmittance, U ($W/m^2K$), of thermal envelope elements according to different energy demand levels.

| Thermal Envelope Element | L1 [1] | L2 [2] | L3 [3] |
|:---:|:---:|:---:|:---:|
| Facades | 0.82 | 0.56 | 0.38 |
| Roofs | 0.45 | 0.44 | 0.33 |
| Windows | 3.3 | 2.3 | 2 |

[1] CTE DBHE-2006. Values according to Table 2.2.-HE1 for climate zone B3. [2] CTE DBHE-2019. Values according to Table 3.1.1a-HE1 for climate zone B. [3] CTE DBHE- 2019. Values according to Table a-Annex and HE for climate zone B.

In total, 21 sets of improvement measures are proposed for the energy renovation strategy, according to each building type, combining three energy demand levels (L1, L2 and L3) and seven sets of improvement measures for thermal envelope elements (Table 4).

**Table 4.** Set of measures for thermal envelope according to a combination of energy demand levels and elements in thermal envelope to be renovated.

| Thermal Envelope Element | L1 | L2 | L3 |
|:---:|:---:|:---:|:---:|
| Windows (W) | L1 W | L2 W | L3 W |
| Facades (F) | L1 F | L2 F | L3 F |
| Roofs (R) | L1 R | L2 R | L3 R |
| Windows + Facades (WF) | L1 WF | L2 WF | L3 WF |
| Windows + Roofs (WR) | L1 WR | L2 WR | L3 WR |
| Facades + Roofs (FR) | L1 FR | L2 FR | L3 FR |
| Windows + Facades + Roofs (WFR) | L1 WFR | L2 WFR | L3 WFR |

The improvement measures proposed for representative school buildings, type A and B, whose geometry and facade design cannot be altered, are found in facades, introducing inner insulation plasterboard lining with metal framing, and on the inside of sloping roofs in type A buildings, through a removable plaster false ceiling with thermal insulation. Both for facades and flat roofs in type C buildings, different exterior thermal insulation systems are proposed.

The type of insulation and the thickness used for facades and roofs, as well as the characteristics of glazing are different, depending on the transmittance to be obtained.

### 2.1.3. Calculation of Global Costs of Sets of Measures

The calculation of global costs of sets of measures is made for a period of 30 years, as established by the Delegated Regulation 244/2012/EU for public buildings.

The global costs include those related to the intervention itself, building consumption and maintenance during the calculation period. Following the regulation, for cost calculation of each set of measures, the initial investment is considered ($C_I$), as well as replacement costs, disposal costs, annual energy costs and annual rise in energy price, the annual maintenance cost of measures, as well as the residual value of elements added.

The Equation (1) used for global cost calculation is:

$$C_g\,(\tau) = C_I + \sum_j \left[\sum_{i=1\,(\tau)} (C_{a,i}\,(j) \times Rd\,(i)) - V_{f,\tau}\,(j)\right] \tag{1}$$

In which: $\tau$ indicates the calculation period; $C_g\,(\tau)$ indicates global cost (referred to starting year $\tau_0$) over the calculation period; CI indicates initial investment costs for implementing measure or set of measures j; $C_{a,i}\,(j)$ indicates the cost per year, i for measure or set of measures, j; $V_{f,\tau}\,(j)$ indicates residual value of measure or set of measures j at the end of the calculation period (discounted of the starting year $\tau_{0)}$; and Rd (i) indicates discount factor for year i.

Consequently, the optimal cost of measures or sets of measures would be that with the lowest energy consumption at the lowest global cost per m² and year. As an example, to facilitate understanding and subsequent interpretation of global cost graphs, a graph is included (Figure 3). It shows the resulting cost-optimal curve (orange line), the initial state of the building (L0), the global cost in the initial state (red line) and the optimal-cost sets of measures (green line).

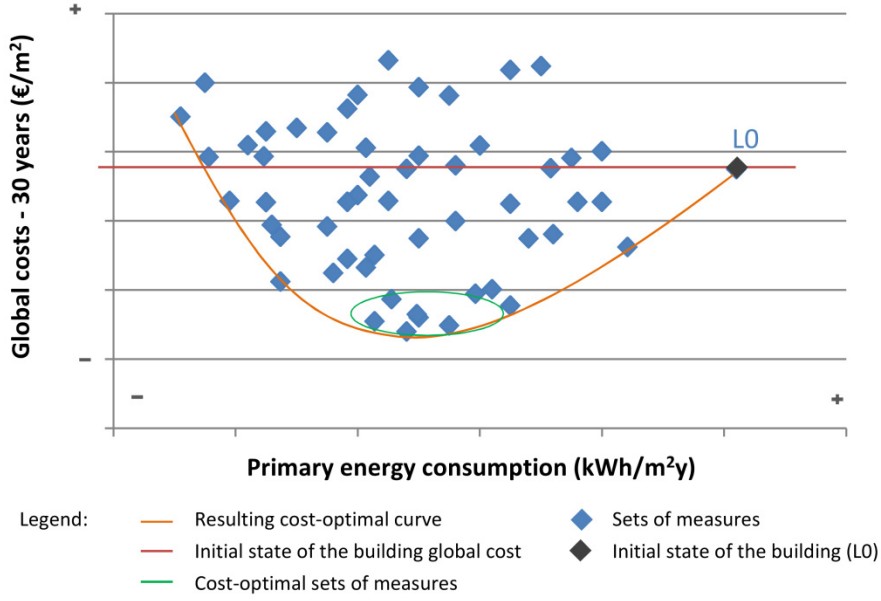

**Figure 3.** Example of cost-optimal graph. Correlation between the 30-year global costs of sets of measures proposed per m², and building's primary energy consumption per m² and year.

The sets of measures with global cost above the optimal cost in the initial state (L0) have a period to recover initial investment longer than the calculation period itself, in this case over 30 years.

### 2.2. Implementing Legislation

The Spanish regulation on energy efficiency, the Technical Building Code DB-Energy Saving (CTE DB-HE) [22] contains basic requirements on the matter, which have been modified in accordance with the directives and goals set by the EU.

Currently, regarding energy demand levels for energy saving, the CTE DB-HE establishes maximum transmittance ($U_{lim}$) for thermal envelope elements, both new and renovated, and a global ratio of heat transmission through thermal envelope ($K_{lim}$). It also limits consumption of non-renewable primary energy ($C_{ep,nren}$) and total primary energy consumption ($C_{ep,tot}$).

On the other hand, concerning renovation of specific buildings, the regulation allows greater flexibility in obtaining some values required, as in the case of buildings with significant architectural value. In the same way, it classifies interventions into small or large ones, and sets limits or requirements according to the type of performance.

This study also analyses requirements for each building type, and the level of compliance with regulation, based on the sets of measures to be implemented.

## 3. Discussion on Results

*3.1. Results on Global Costs of Measures and Primary Energy Consumption*

It should be emphasised that each school building is studied in the initial state at an energy demand level, L0, subsequently implementing sets of measures, as seen in Sections 2.1 and 2.1.2.

As a result of relating 30-year global costs per m$^2$ to each set of measures and the consumption of primary energy per m$^2$ and year, the graphs obtained show an optimal intervention cost for each school building type (Figure 4a–c).

Overall, it is noted that, in the three building types, for the same combination of measures, as in the WF thermal envelope elements, the energy demand levels (L1, L2 and L3) do not imply great differences in terms of global costs and resulting energy consumption. The graph clearly shows how different combinations of measures are grouped together according to construction elements enhanced.

Moreover, when comparing the graphs of representative building types it is clear what set of measures or thermal envelope elements obtains optimal costs.

For type A buildings, sets of measures introducing roof insulation obtain a greater reduction in energy consumption at the lowest global cost over 30 years. In this case, and depending on the energy demand level in terms of transmittance, consumption reduction regarding the initial state (L0) is between 24.5% and 25.3%, for interventions on the roof (R); between 28.1% and 30.2%, for improvements in facade and roof (FR); between 30.8% and 32.2%, for windows and roofs (WR); and between 34.8% and 38.1%, if the three thermal envelope elements (WFR) undergo improvements. The energy saving gained would enable to recover initial investment in such improvements within 4–5 years (R), 5–6 years (FR), 15 years (WR) and 14 years (WFR). The cost-optimal levels of interventions would be R and FR, thermal envelope elements, with a lower global cost. A greater saving is obtained through the OR and OFR, elements improved in the thermal envelope.

For type B and C buildings respectively, the sets of measures with the lowest global cost over 30 years include facade thermal insulation (F), and facade and roof thermal insulation (FR). These sets of measures represent a decrease in consumption between 15.8–18.6% (F) and 16.6–19.2% (FR) for type B buildings, and between 18.6–21.1% (F) and 20.5–22.9% (FR) for type C buildings. The repayment term is less than 7 years in type B buildings, and between 12–13 years (R) and 19 years (FR) in type C buildings.

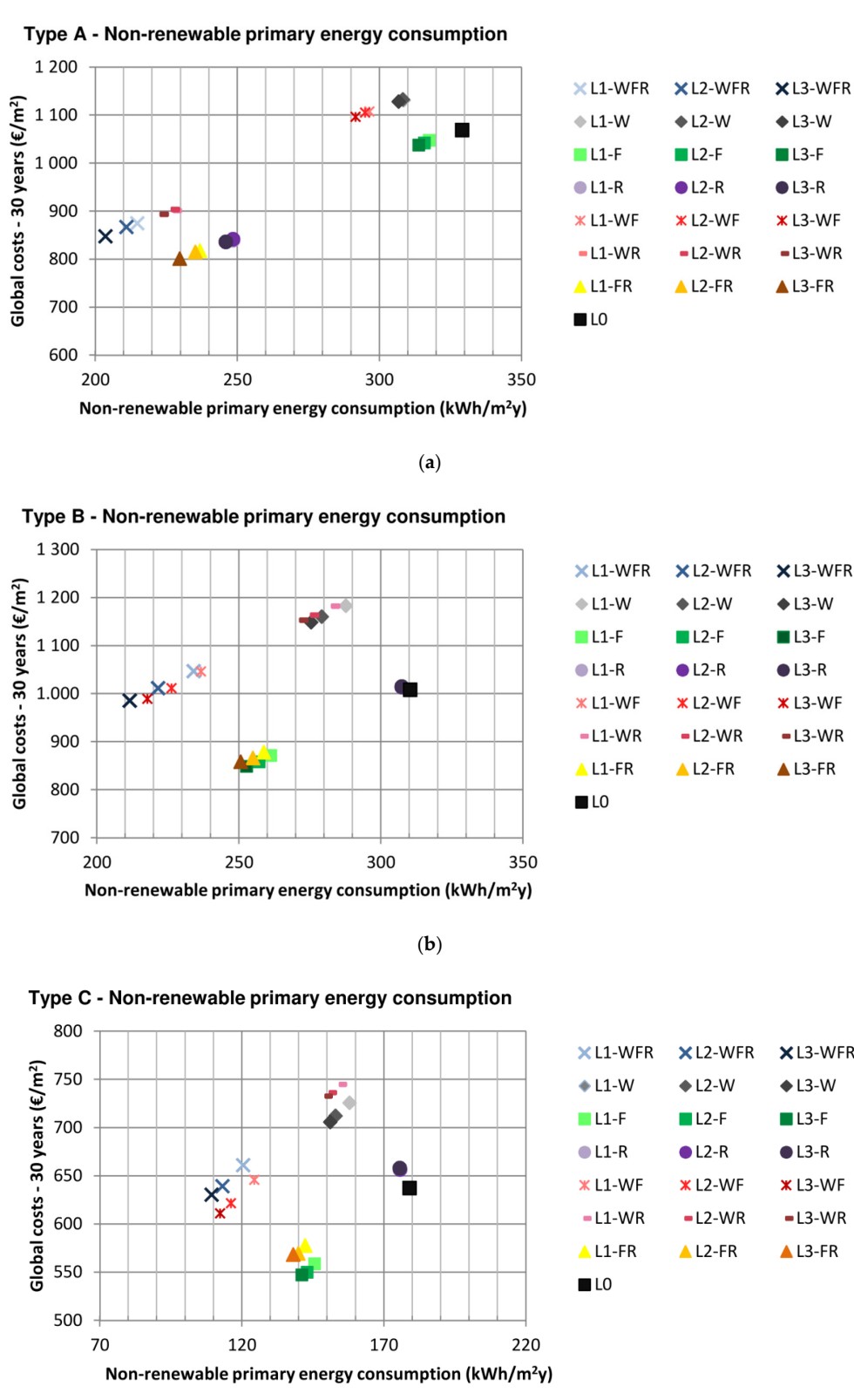

**Figure 4.** Cost-optimal graphs. Correlation between 30-year global cost per m$^2$ in the sets of measures proposed and building's primary energy consumption per m$^2$ and year, after the intervention, assuming a price increase rate of 1%. (**a**) Type A school building; (**b**) type B school building; (**c**) type C school building.

In building types B and C, a greater reduction in energy consumption is obtained after replacing facades and windows (WF), as well as renovating all thermal envelope elements (WFR). In both sets of measures, the global costs increase, which means a longer repayment term for the investment made. In type B buildings, the global cost would be very similar to 30-year global cost, without necessary performance in the building. That is, the initial state L0, even exceeding it in some cases, which implies that the 30-year investment recovery is greater.

Table 5 is drawn up in order to provide accurate figures and facilitate comparison with some results included in the previous graphs. It shows 30-year global costs per m$^2$ corresponding to the most relevant sets of measures which include the full amount to pay for energy consumption in 30 years for each m$^2$ (living area), the implementation of measures taken and maintenance. It also includes global cost per m$^2$ after 30 years of the initial state (L0), that is, if no action is taken, and identifies the full amount to pay for each m$^2$, especially energy consumption expenses.

**Table 5.** 30-year global cost of initial state L0 per m$^2$, according to type of school building, and global cost of sets of measures per m$^2$, including those with better energy performance.

| Type of School Building | L0 | Elements Improved | L1 | L2 | L3 |
|---|---|---|---|---|---|
| Type A | €1069.29 | R | €840 | €842 | €836 |
| | | FR | €818 | €815 | €802 |
| | | WR | €901 | €904 | €894 |
| | | WFR | €875 | €868 | €848 |
| Type B | €1008.40 | F | €871 | €858 | €849 |
| | | FR | €879 | €866 | €859 |
| | | WF | €1047 | €1012 | €990 |
| | | WFR | €1047 | €1012 | €986 |
| Type C | €637.46 | F | €559 | €533 | €547 |
| | | FR | €578 | €570 | €568 |
| | | WF | €646 | €622 | €611 |
| | | WFR | €661 | €639 | €631 |

The fact that the global cost of specific sets of measures is higher than that in the initial state entails that the starting investment is not amortised in 30 years. These sets of measures could be implemented together with an enhancement of building's facilities, namely, by taking measures to be implemented in thermal installations. In this way, global costs would be minimised through a reduction in energy consumption due to improvement in system performance.

The annual savings generated after implementing sets of measures concerning energy renovation in each building can be expressed from the viewpoint of emission reduction (Table 6), and the subsequent economic saving after minimizing annual energy consumption per year (Table 7). Table 8 shows the reduction percentage in energy consumption and recovery period of initial amortisation for sets of improvement measures with lower 30-year global cost and a greater consumption reduction, according to building type.

**Table 6.** Emission reduction compared to the initial state per year, according to type of school building and sets of measures. Expressed in $KgCO_2$ per year and according to building.

| Type of School Building | Elements Improved | L1 | L2 | L3 |
|---|---|---|---|---|
| Type A | R | 10,766.4 | 10,766.4 | 11,080.8 |
| | FR | 12,338.2 | 12,573.9 | 13,281.2 |
| | WR | 13,517.0 | 13,595.6 | 14,145.7 |
| | WFR | 15,245.9 | 15,796.0 | 16,739.0 |
| Type B | F | 4413.5 | 4781.3 | 5149.1 |
| | FR | 4623.7 | 4938.9 | 5306.7 |
| | WF | 6410.1 | 7513.5 | 8249.1 |
| | WFR | 6830.5 | 7933.8 | 8827.1 |
| Type C | F | 17,740.1 | 19,135.4 | 20,132.0 |
| | FR | 19,534.0 | 20,730.0 | 21,726.6 |
| | WF | 28,503.8 | 32,291.0 | 34,483.6 |
| | WFR | 31,095.0 | 34,682.9 | 36,875.5 |

**Table 7.** Average annual saving according to type of school building due to reduction in energy consumption, according to sets of measures.

| Type of School Building | Elements Improved | L1 | L2 | L3 |
|---|---|---|---|---|
| Type A | R | €6868.81 | €6868.81 | €7078.37 |
| | FR | €7858.39 | €7991.44 | €8458.78 |
| | WR | €8630.09 | €8693.29 | €9022.59 |
| | WFR | €9732.76 | €10,057.07 | €10,684.08 |
| Type B | F | €2799.90 | €3032.30 | €3284.72 |
| | FR | €2936.67 | €3153.51 | €3399.25 |
| | WF | €4080.88 | €4784.74 | €5266.22 |
| | WFR | €4343.30 | €5053.84 | €5624.27 |
| Type C | F | €8297.11 | €8954.29 | €9410.83 |
| | FR | €9133.15 | €9741.87 | €10,192.79 |
| | WF | €13,510.74 | €15,446.24 | €16,439.34 |
| | WFR | €14,599.98 | €16,362.83 | €17,343.28 |

**Table 8.** Reduction percentage in energy consumption and recovery period of initial amortisation in sets of measures with lower 30-year global cost and a greater consumption reduction, according to type of school building.

| Type of School Building | Elements Improved | % Reduction in Energy Consumption | | | Amortisation of Investment (Years) | | |
|---|---|---|---|---|---|---|---|
| | | L1 | L2 | L3 | L1 | L2 | L3 |
| Type A | R | 24.5% | 24.5% | 25.3% | 4 | 5 | 5 |
| | FR | 28.1% | 28.5% | 30.2% | 5 | 6 | 6 |
| | WR | 30.8% | 31.0% | 32.2% | 15 | 15 | 15 |
| | WFR | 34.8% | 35.9% | 38.1% | 14 | 14 | 14 |
| Type B | F | 15.8% | 17.1% | 18.6% | 5 | 4 | 5 |
| | FR | 16.6% | 17.8% | 19.2% | 7 | 7 | 7 |
| | WF | 23.1% | 27.1% | 29.8% | >30 | >30 | 29 |
| | WFR | 24.6% | 28.6% | 31.8% | >30 | >30 | 29 |
| Type C | F | 18.6% | 20.1% | 21.1% | 13 | 12 | 13 |
| | FR | 20.5% | 21.9% | 22.9% | 19 | 19 | 19 |
| | WF | 30.5% | 35.1% | 27.3% | >30 | 29 | 28 |
| | WFR | 32.8% | 36.8% | 38.9% | >30 | >30 | 30 |

If the results obtained for each building type were implemented in all representative schools, the result obtained by applying the lowest overall cost measures N3 FR (Type

A) and N3 F (Types B and C) would be an average saving of 50,752.68 euros per year, 37,391.75 euros and 295,590.91 euros, respectively.

In addition, annual emission reductions would be 79,687.2 $KgCO_2$, 56,640.1 $KgCO_2$ and 583,828 $KgCO_2$, respectively.

### 3.2. Regulatory Requirements and Compliance

Regarding Spanish Regulations on energy saving and compliance when it comes to building renovation, the CTE DB-HE 2019 [22] sets a global energy consumption limitation (non-renewable consumption of primary energy ($C_{ep,nren}$) and primary energy total consumption ($C_{ep,tot}$)) when there are interventions in over 25% of thermal envelope, as well as in thermal installations.

Specifically, within the climate zone corresponding to the city of Valencia, the regulatory limit value established for the $C_{ep,nren}$ is 55 kWh/m2 per year, and 80 kWh/$m^2$ per year for the $C_{ep,tot}$.

In the case of the school buildings studied, the $C_{ep,nren}$ value in the initial state of type A buildings (L0) is 351.3 kWh/$m^2$ per year, 256.9 kWh/$m^2$ per year for type B buildings and 192.1 kWh/$m^2$ per year for type C buildings. The greatest reduction in energy consumption would be obtained through the set of measures L3-WFR. In particular, this set of measures would allow the $C_{ep,nren}$ values to be 207.6 kWh/$m^2$ per year for type A buildings, 183.1 kWh/$m^2$ per year for type B and 116.3 kWh/$m^2$ per year for type C.

Regarding $C_{ep,tot}$, its value in the initial state for (L0) type A buildings is 425.7 kWh/$m^2$ per year, 311.3 kWh/$m^2$ per year for type B and 205.3 kWh/$m^2$ per year for type C. The set of measures L3-WFR would allow $C_{ep,tot}$ values to be reduced in 251.6 kWh/$m^2$ per year for type A buildings, 221.9 kWh/$m^2$ per year for type B and 128.8 kWh/$m^2$ per year for type C.

As mentioned above, this study is focused on the renovation of thermal envelope elements, so this limitation is not mandatory. However, it should be considered in the case of performing in facilities.

Below, the study analyses the compliance with requirements on transmittance of thermal envelope elements.

In any intervention on thermal envelope, every single element renovated must meet requirements on transmittance set by regulations, for example, in the case of window replacement. These values partly correspond to those set for level L2 in Table 2. Level L1, as indicated in Section 2, does not meet the CTE DB-HE 2019 regulation.

Particularly, for major renovations over 25% of thermal envelope, the standard also sets a global heat transmission ratio (K). This depends on the climate zone where buildings are located and their compactness. According to the CTE DB HE, the K coefficient indicates the average value of heat transfer ratio for heat exchange surface of the thermal envelope ($A_{int}$).

As shown in Table 9, the $K_{lim}$ level set in the regulations is reached only by implementing one set of measures (L3-WFR) in type C buildings. Consequently, to reduce global ratio of the sets of measures proposed, it is necessary to minimise transmittance, that is, by setting more restrictive energy demand levels in facades (F), roofs (R) and windows (W). Another way to minimise building's energy demand is by performing in other elements, such as interior partitions and floors in contact with unheated rooms. This is not always technically or economically feasible.

**Table 9.** Global heat transmission ratio K (W/m$^2$K). Regulatory requirements according to type of school building, building compactness and climate zone, Klim and ratios obtained for each set of measures and energy demand level.

| Type of School Building | Klim | Elements Improved | K (W/m$^2$K) | | |
|---|---|---|---|---|---|
| | | | L1 | L2 | L3 |
| Type A | 0.9 | R | 2.83 | 2.83 | 2.83 |
| | | FR | 2.25 | 2.07 | 1.91 |
| | | WR | 2.44 | 2.19 | 2.13 |
| | | WFR | 1.86 | 1.43 | 1.21 |
| Type B | 0.85 | F | 2.27 | 2.18 | 2.09 |
| | | FR | 1.94 | 1.84 | 1.73 |
| | | WF | 1.90 | 1.62 | 1.46 |
| | | WFR | 1.57 | 1.29 | 1.11 |
| Type C | 0.86 | F | 2.31 | 2.21 | 2.14 |
| | | FR | 1.72 | 1.62 | 1.42 |
| | | WF | 1.82 | 1.54 | 1.50 |
| | | WFR | 1.23 | 0.95 | 0.78 |

In view of this situation, legislation provides for the fact that occasionally it is not possible to reach the level of benefit generally established. In these cases, some solutions may be adopted to achieve the highest level of adequacy. This is possible, as long as they are buildings with recognised historical or architectural value, when other solutions are not technically or economically feasible, or solutions involve substantial changes in elements of thermal envelope, or in thermal facilities without initial intervention.

For this study, type A and B school buildings could benefit from this flexibility criterion included in the standard, provided that measures implemented are close to limit values.

Regarding solar control, in the case of renovating over 25% of the total area of the final thermal envelope, the solar control parameter (q$_{sol;jul}$) should not exceed the limit value 4 kWh/m$^2$ per month, for uses other than private and residential.

According to the CTE DB HE, q$_{sol;jul}$ indicates the ratio between the solar gains of windows and doors on the thermal envelope during the month of July with mobile solar protection activated, and the useful floor area included in spaces within the thermal envelope (A$_{useful}$). Solar protection can be implemented in all the building or in part of it.

This solar control parameter is fulfilled in those sets of measures in which windows and doors (O) are replaced, since protection elements are included. In type A and B buildings, as it is not possible to alter facades, protection systems should be installed on the inside, for example by using shutters.

## 4. Conclusions

Through the implementation of optimum-cost methodology, the study aims at identifying the set of improvement measures with optimum intervention cost for each building type studied. It is found that for type A buildings, the most cost-effective intervention would be to renovate facades and roofs (L3-FR), whereas for types B and C, the improvement is only made in facades (L3-F). In all cases, the set of improvement measures that brings buildings closer to nZEB involves performing in all thermal envelope elements with the most restrictive values of thermal transmittance (L3-WFR).

All in all, the implementation of cost-optimal sets of improvement measures in 46 schools studied results in an average annual saving of 378,735.34 euros and an annual emission reduction of 720,155.3 KgCO$_2$.

These results show Public Administration in Valencia that the most cost-effective solution is not always the same for all schools, and depends on building typology. This suggests that it is not advisable to renovate the same thermal envelope element simultaneously in all buildings for instance, by renovating windows in all schools. In order to save costs when performing in more than one element or school at the same time, the

typology classification proposed for the sample is suggested to be applied so that the same intervention is replicated for the same typology but not for all schools.

The results achieved allow school principals in Valencia to make decisions on carrying out priority renovations in buildings and potential strategies. For example, for those measures with a global cost similar or higher than that in the current state of the building, it is appropriate to perform a deep renovation, including enhancement of thermal envelope and heating and cooling systems, so that the 30-year global cost is reduced. The long-term cost reduction and the quick return of investment of some sets of measures may be a reason for public administrations in Valencia to invest in school renovation.

Regarding regulatory requirements, it is shown that an approach to building renovation with the aim of complying with current legislation for new buildings may not be the most cost-effective option.

In summary, the cost-optimal methodology applied to the renovation of school buildings studied provides quantitative data on costs and energy saving that can be obtained after implementing specific sets of measures in 46 school buildings in the city of Valencia. All this facilitates to identify, among other data, the initial cost of measures, consumption reduction gains and the return on investment periods. These data can provide public administrations in Valencia with criteria to design long-term intervention plans, which enable available resources to be efficiently invested.

Irrespective of the specific results reached, the adaptation of the methodology proposed to a wider scale (regional/national) can help to build support for deciding about the renovation of school buildings and designing long-term renovation strategies. The study can also be expanded to other buildings in the city, as well as other cities or regions.

**Author Contributions:** Conceptualization, M.E.L.-D.; Methodology, B.S.-L. and L.O.-M.; Writing—original draft, M.E.L.-D.; Writing—review & editing, M.E.L.-D., B.S.-L. and L.O.-M. All authors have read and agreed to the published version of the manuscript.

**Funding:** This research received no external funding.

**Institutional Review Board Statement:** Not applicable.

**Informed Consent Statement:** Not applicable.

**Conflicts of Interest:** The authors declare no conflict of interest.

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
