# Peer review of "Identification of Cost-Optimal Measures for Energy Renovation of Thermal Envelopes in Different Types of Public School Buildings in the City of Valencia"

_applsci, doi:10.3390/app11115108_

Round 1

Reviewer 1 Report

The authors present a summary of cost optimal studies for school buildings in Spain. 

Whilst the topic per se is relevant, the paper mostly lacks scientific detail and originality and subsequently does present few new findings that go beyond similar studies that have been published by the individual member states (see EU countries cost optimal reports on ec.europe.eu). 

The abstract is poorly written and does not provide the relevant details. 

The introduction provides an adequate overview of the relevant legislative context. The State of the art (which should not be part of the introduction) is however rather short and more a summery of sources rather than a critical review of the current state of the art related to the topic of the article. Sources are highly summarised (e.g. on page 2 in the last paragraph 10 sources are cited within 3 lines), thus not providing enough context. 

The methodology and results would benefit greatly from coherent english editing. It parts the meaning is difficult to understand due to poor translation (see below comments on english). 

The conclusions are rather simplistically written, mostly stating the obvious. This should be completely revised highlighting the specific findings of the study. 

The english language needs significant revision, in parts the text sounds like an automatic translation. Some of the relevant wording (e.g. "opening") lacks the correct translation (in this context this would actually mean "transparent element" or "window"). 

Overall the article would be more suitable for a conference paper than a scientific journal. If it is resubmitted it should be extensively revised. 

Author Response

Point 1: The abstract is poorly written and does not provide the relevant details. 

Response 1: We have changed it.

Point 2: The introduction provides an adequate overview of the relevant legislative context. The State of the art (which should not be part of the introduction) is however rather short and more a summary of sources rather than a critical review of the current state of the art related to the topic of the article. Sources are highly summarised (e.g. on page 2 in the last paragraph 10 sources are cited within 3 lines), thus not providing enough context. 

Response 2: We have changed it. We have simplified the content by trying to talk only about what is related to the main topic of the article, without going into other aspects related to buildings or their requirements.

Point 3: The conclusions are rather simplistically written, mostly stating the obvious. This should be completely revised highlighting the specific findings of the study. 

Response 3: We have rewritten them

Point 4. The methodology and results would benefit greatly from coherent english editing. It parts the meaning is difficult to understand due to poor translation (see below comments on english).

Response 4: We have made a revision.

Reviewer 2 Report

Article shows case study of cost evaluation of measures for improving efficiency of thermal envelope of three types of school buildings. From methodology, point of view paper does not bring new knowledge to the reader. It is not shown how EP,tot is determined as delivered energy and energy carriers are not show. The content of paper repeats several time and initial conditions should be presented in introduction. (like Klim, national required values of nZEB, ...). Some results presented duplicates in Fig. and Table, which is not needed. Some indicators are not explained in discussion, for example qsol.

Therefore, I cannot recommend the paper to be published. Some remarks are presented bellow.

Abstract must content presentation of findings presented in the paper.

Page 2 paragraph 2 meaning in not clear, can be deleted.

Page 3 paragraph 3 “the whole” ? schools building sector ?

Page 3 paragraph 7, not bring new issues, sourly this is mentioned in cited literature before this paragraph. (as well as paragraph 8)

TOOLS ?

Page 3, chapter 2, point 4 citation is needed

How type A and B differ ? Type is not mentioned

Thermal transmittance U

Openings = windows ?

Initial state N0 of L0 ? (Table 1)

2.1.2. Are this passive renovation measures of improving thermal envelope mentioned before ?

2.1.2. paragraph 2, schools are already presented in previous chapter

2.1.2. paragraph 7, this should be presented introduction to emphases the problem that will be presented

Page 5 paragraph 1 21 measures or measurements ?

Table 3 what are O, F not stated (openings ,,,,,,)

2.1.3. paragraph 1 already mentioned

2.2. already explained

2.2. paragraph 2, Ep,tot (EN 52000-1) ? National requirements must be shown earlier in text

Figure 2 is not needed, reference is cited, arrow NZEB is nZEB ? which value, defer to much from minimum cost ? 

Fig. 3 Ep,tot or Ep,nres ?

Price increase for fossil flues and electricity are the same ?

Values in Table 4 cannot be determine without Table 5 ?

Values in Table 5 are shown in Fig. 3 aren’t they, Which additional information shows Table 5 ?

Table 7 must be before Table 4 (amortization period)

It is strange that limit values re shown after cost evaluation (Table 8) and after environment indicators (Table 6)

Page 11, qsol, there is no evidence how is determined ?

Author Response

Response to Reviewer 2 Comments

Article shows case study of cost evaluation of measures for improving efficiency of thermal envelope of three types of school buildings. From methodology, point of view paper does not bring new knowledge to the reader. It is not shown how EP,tot is determined as delivered energy and energy carriers are not show. The content of paper repeats several time and initial conditions should be presented in introduction. (like Klim, national required values of nZEB, ...). Some results presented duplicates in Fig. and Table, which is not needed. Some indicators are not explained in discussion, for example qsol.

POINT 1: Different remarks

- Abstract must content presentation of findings presented in the paper. We have rewritten it

- Page 2 paragraph 2 meaning in not clear, can be deleted. We have rewritten the introduction

- Page 3 paragraph 3 “the whole” ? schools building sector ? We have explain it better

- Page 3 paragraph 7, not bring new issues, sourly this is mentioned in cited literature before this paragraph. (as well as paragraph 8) (Page 4)Yes, this normative has been cited before but as we have use different normative revisions, we have put here the specific references [].

- TOOLS ? We have used an Spanish software that is an energy calculator,  page 3, point 2.1. Just before point 2.1.1.

For global cost calculation we have use a tool  (system) that IVE (Instituto Valenciano de la Edificiación) has developed. This clarification has been added  to the article.

- Page 3, chapter 2, point 4 citation is needed. (corrected)

How type A and B differ? Type is not mentioned.  We have ad a new table (Table 1) with the main characteristics of each type

- Thermal transmittance U – Corrected

- Openings = windows? (Translation mistake) In spanish there is a word that includes all the windows and doors. In order to a better comprehension of the article we have decide to use “windows”.

- Initial state N0 of L0 ? (Table 1) (Translation mistake) Corrected

- 2.1.2. Are this passive renovation measures of improving thermal envelope mentioned before? No, we are explaining them now

- 2.1.2. paragraph 2, schools are already presented in previous chapter. In this paragraph we are trying to explain the problem that exists about the different buildings of each school and their systems.

- 2.1.2. paragraph 7, this should be presented introduction to emphases the problem that will be presented We have added a short mention in the introduction

- Page 5 paragraph 1 21 measures or measurements ? Measures (Corrected)

- Table 3 what are O, F not stated (openings ,,,,,,) – First column includes all the abreviations (now “Table 4”)

- 2.1.3. paragraph 1 already mentioned - We have added more information, in order to explain it better

- 2.2. already explained. - 2.2. paragraph 2, Ep,tot (EN 52000-1) ? National requirements must be shown earlier in text. It’s the first time that we explain national requirements. In a first version of the article we had explained it earlier, but some people who read it, sugest us to explain separately cost optimal information and normative in order to a better comprenhension.

- Figure 2 is not needed, reference is cited, arrow NZEB is nZEB ? which value, defer to much from minimum cost? In a first version of the article we didn´t include this figure, but some people who read it, have several doubts about figures 3 and result. So we decide to introduce in order to a better comprehension.

- Fig. 3 Ep,tot or Ep,nres? Ep,nres (Corrected)

- Values in Table 4 cannot be determine without Table 5? We have change the order

-  Values in Table 5 are shown in Fig. 3 aren’t they, Which additional information shows Table 5?. It includes only main measures results. This table have been included in order to facilitate results comparison and exact quantitative data that the graph does not provide.

- Table 7 must be before Table 4 (amortization period) We have changed it

- It is strange that limit values re shown after cost evaluation (Table 8) and after environment indicators (Table 6). In the study, the thermal transmittance values set by the regulations (L2 and L3) have been used and it has been proposed to intervene in three elements of the envelope. This section includes regulatory compliance in order to see whether it is possible to comply with the regulations with these measures or whether it is necessary to intervene in other elements.

- Page 11, qsol, there is no evidence how is determined? The Spanish regulations establish qsol calculation method. We have used a software that provides results directly.

Reviewer 3 Report

The article deals with an interesting though not new, topic. It is very well written and in a consequential way representative of an excellent scientific method. The development is very clear and rigorous, even if for many aspects related to the specific legislation and construction practice, the conclusions may seem obvious.
The interesting part, more than the analysis of individual buildings, is the possibility that the method takes shape as a decision support system for the large-scale renovation of similar buildings. I would enhance this aspect more in the conclusions and include some reflection on the "savings potential" generated by evaluating the extent of potentially renewable buildings. (also roughly and approximately)

  • It would be useful to highlight a short list of differences between the three school buildings in terms of year of construction (generically before 1979) and materials. Only thermal transmittance is declared but not the materials or layers originally used.
  • Check the phrase on page 5 just before par. 2.1.3:  "the transmittance to be obtain(ed)"
  • Par. 2.2: par 2.2: all abbreviations could be written using subscripts to make the text more understandable. It is clear by the context, but, for example, "Cep, nrem" could be clearly written as "Cep, nrem"; otherwise, it could be interpreted as 2 different abbreviations. "nrem" is correct or there's a typo "nren" (non-renewable)?
  • Page 8 first paragraph: declare HF abbreviation. It appears only on this page, and it is not clear what it stands for.

Author Response

Point 1: The article deals with an interesting though not new, topic. It is very well written and in a consequential way representative of an excellent scientific method. The development is very clear and rigorous, even if for many aspects related to the specific legislation and construction practice, the conclusions may seem obvious.

Response 1: We have rewritten conclusions

Point 2: The interesting part, more than the analysis of individual buildings, is the possibility that the method takes shape as a decision support system for the large-scale renovation of similar buildings. I would enhance this aspect more in the conclusions and include some reflection on the "savings potential" generated by evaluating the extent of potentially renewable buildings. (also roughly and approximately)

Response 2: We have added it

Point 3: It would be useful to highlight a short list of differences between the three school buildings in terms of year of construction (generically before 1979) and materials. Only thermal transmittance is declared but not the materials or layers originally used.

Response 3: We have included a new Table with this information (Table 1)

Point 4: Different remarks

- Check the phrase on page 5 just before par. 2.1.3:  "the transmittance to be obtain(ed)" (Corrected)

- Par. 2.2: par 2.2: all abbreviations could be written using subscripts to make the text more understandable. It is clear by the context, but, for example, "Cep, nrem" could be clearly written as "Cep, nrem"; otherwise, it could be interpreted as 2 different abbreviations. (Corrected)

- "nrem" is correct or there's a typo "nren" (non-renewable)? It was a mistake (Corrected)

- Page 8 first paragraph: declare HF abbreviation. It appears only on this page, and it is not clear what it stands for. It was a mistake (Corrected)

Reviewer 4 Report

Research topics are essential for future sustainable development and deserve attention. In general, the research objectives, methodology, and results of the study are well relevant and may be considered for publication. However, before being accepted, some of the following essential information needs to be more studied and supplemented:
1. Introduction: the introduction of the article is quite lengthy and does not clarify the necessity of the study. The author should clarify the role and necessity of the study in this section. In addition, it should also be summarized.
The literature review section also needs to be supplemented, it is necessary to clarify studies with the same topics in the past. Considering the work, methods, and results that previous studies have done to find out the gaps that need to be solved and improved the method of this study.
2. Research Methods: As mentioned above, the lack of literature review leads to an uncertain method choice. Specifically, there are many methods to evaluate and quantify building improvement strategies such as LCA, LCC,... The author needs to clarify the reasons for choosing the method that is being used in this study but not another.
3. Conclusion: The conclusion has clarified the results of the study, but should point out the limitations of the study (number of projects, assumptions, the scope of the study ...). Since then, proposing researches that need to be done in the future.

Author Response

Point 1, Introduction: the introduction of the article is quite lengthy and does not clarify the necessity of the study. The author should clarify the role and necessity of the study in this section. In addition, it should also be summarized.

The literature review section also needs to be supplemented, it is necessary to clarify studies with the same topics in the past. Considering the work, methods, and results that previous studies have done to find out the gaps that need to be solved and improved the method of this study.

Response 1: We have rewritten some parts

Point 2, Research Methods: As mentioned above, the lack of literature review leads to an uncertain method choice. Specifically, there are many methods to evaluate and quantify building improvement strategies such as LCA, LCC,... The author needs to clarify the reasons for choosing the method that is being used in this study but not another.

Response 2: We have tried to explain better in introduction.

Point 3, Conclusion: The conclusion has clarified the results of the study, but should point out the limitations of the study (number of projects, assumptions, the scope of the study ...). Since then, proposing researches that need to be done in the future.

Response 3: We have included more information about that.

Round 2

Reviewer 1 Report

The authors have somehow improved the paper, however there are still some significant weaknesses.

The abstract is still poorly written and not concise in its meaning. It should contain (1) the context and societal relevance, (2) the method, (3) the key results. Already the first sentence of the abstract is both unclear from an english language perspective and from its meaning. 

The state of the art (see comment from first review) is still not adequately addressed. The authors have in response to the previous comment simply taken out over 15 references. This is not an improvement and subsequently an adequate discussion on the state of the art is still mostly missing. 

The conclusions are (like the abstract) still poorly written and not very meaningful. To start the conclusions with "some of the conclusions are included in the results of the study" is in itself rather meaningless. The second sentence (..."scarcity on data on school buildings...") and following conclusions should clearly highlight that the study is only focused on a very narrow data range (i.e. school buildings in Valencia). Therefore the statements made should be clearly focused on this data-range only (i.e. some of the sentences suggest a general answer to the topic of cost-optimality in school buildings, which the study clearly does not provide).

Similarly the title is rather misleading (Mediterranean climate) as this would suggest a much broader approach and data range than the results the study actually provides. 

In summary, whilst the study itself is not un-interesting, it suggests from its text much broader conclusions than what can be actually derived from the analysed data. This is not good scientific practice and should therefore be amended with much more care for the details and the actual limited conclusions that can be drawn from this study. 

As noted before, the english and overall syntax should be extensively revised. Whilst some of the text has been improved, there are still a series of sentences that are exceptionally poorly written. The paper would greatly benefit from a revision by a native speaker. 

On a general note: it would be very helpful to use a template with line numberings and to subsequently also refer to the changes with the respective line numberings. Also, to respond to each comment with "we have changed it" is not really helpful for the reviewers in order to identify the changes made. 

Author Response

Point 1: The abstract is still poorly written and not concise in its meaning. It should contain (1) the context and societal relevance, (2) the method, (3) the key results. Already the first sentence of the abstract is both unclear from an english language perspective and from its meaning. 

Response 1: We have rewritten it.

Point 2: The state of the art (see comment from first review) is still not adequately addressed. The authors have in response to the previous comment simply taken out over 15 references. This is not an improvement and subsequently an adequate discussion on the state of the art is still mostly missing.

Response 2: We have added information about the reference.

Point 3: The conclusions are (like the abstract) still poorly written and not very meaningful. To start the conclusions with "some of the conclusions are included in the results of the study" is in itself rather meaningless. The second sentence (..."scarcity on data on school buildings...") and following conclusions should clearly highlight that the study is only focused on a very narrow data range (i.e. school buildings in Valencia). Therefore the statements made should be clearly focused on this data-range only (i.e. some of the sentences suggest a general answer to the topic of cost-optimality in school buildings, which the study clearly does not provide).

Response 3: We have included in different sentences “Valencia schools” or “School studied (marked in blue). We have removed the sentence “some of the conclusions are included in the results of the study"

Point 4: Similarly the title is rather misleading (Mediterranean climate) as this would suggest a much broader approach and data range than the results the study actually provides. 

Response 4: We have changed it. In the title, we have written that the study is about schools of Valencia.

Point 5: In summary, whilst the study itself is not un-interesting, it suggests from its text much broader conclusions than what can be actually derived from the analysed data. This is not good scientific practice and should therefore be amended with much more care for the details and the actual limited conclusions that can be drawn from this study. 

Response 5: We have removed some conclusions and we have tried to reduce globals conclusions about schools.

Changes are marked by colours: 

  • Yellow: main changes revision 1
  • Blue: main changes revision 2

Using the "Track Changes" function it is possible to see translation corrections.

Reviewer 2 Report

Authors improve the article, but it needs further improvements by replying following comments:

Study is performed for “Mediterranean climate” - indicate how such climate is defined (classification)

Abstract 3 paragraph - (“several” mislead reader as it can be understand more then 3), proposal: In the article primary schools in City of Valencia was analyzed and classified into three representative types. For each type 21 improvements.....

As definition of schools type is shown as novelty, the types must be presented with short ‘definition”; later in text this could be type A, B, and C, build some kind of “theoretical classification” is needed.

Table 1 , Type B do not looks like two floor building

Table 2 , please check Uroof, 4.17 W/m2K is quite high if consist 0f close air gap (chamber); which standard was used for determination of U?

Table 2 , what is meaning of two values of “windows” single-glazed (5,7/2,2)?

Figure 2 , If authors insist on this figure, it must be mentioned which study case is shown: we insist that general data [22] is not needed and must be replaced by study case. Is this graph for “education building” ? The limits of nZEB + and - from minimum is not shown (this will represent limit of Ep) that must be required by RU state. As mentioned before, on Fig 2 is not shown if data are Ep,tot or Ep,nren.

Chapter 2.2. Limits of Ep,nren and Ep,tot are mentioned, but I can not find the values in discussion and comments. It will be better to use nomenclature from EN 52000-1 (E instead of C). It will be useful that requirements will be shown in Fig. 3.

As suggested in first review, the cash-flow mathematical equation will be helpful for the readers. The example can be, for example, find in paper (DOI: 10.2495/GD170151).

The article would be interesting world-wide. Because of that, the K and qsol must be explained by physical explanation. This was requested in the previous review as well. What is the difference between solar control parameter and solar factor? (last paragraph in 3.2.)

Author Response

Point 1: Study is performed for “Mediterranean climate” - indicate how such climate is defined (classification)

Response 1: Schools are located in Valencia, that’s why we have been written that are Mediterranean climate schools.

We have changed the title. We have written that the study is about schools of Valencia (Marked in blue).

Point 2: Abstract 3 paragraph - (“several” mislead reader as it can be understand more then 3), proposal: In the article primary schools in City of Valencia was analyzed and classified into three representative types. For each type 21 improvements.....

Response 2: Yes, it can be understand more than 3. We have included you proposal.

Point 3: As definition of schools type is shown as novelty, the types must be presented with short ‘definition”; later in text this could be type A, B, and C, build some kind of “theoretical classification” is needed.

Response 3: We have added a short definition of the types. Chapter 2.1.1.

Point 4: Table 1, Type B do not looks like two floor building

Response 4: Yes, it is a mistake. It’s a three floor building. We have correct it

Point 5: Table 2, please check Uroof, 4.17 W/m2K is quite high if consist 0f close air gap (chamber); which standard was used for determination of U?

Response 5: The gap of this type of buildings is a high ventilated gap. It is considered that the thermal envelope is a false ceiling made of cane and plaster.

The specific values of Uroof had been calculated by Instituto Valenciano de la Edificación in: “Catálogo de soluciones constructivas de rehabilitación” ISBN: 978-84-96602-72-4

Point 6: Table 2, what is meaning of two values of “windows” single-glazed (5,7/2,2)?

Response 6: First value is U of single glazed and the second one is U of windows structutre (wood or metal). We have added this information in the table.

Point 7: Figure 2, If authors insist on this figure, it must be mentioned which study case is shown: we insist that general data [22] is not needed and must be replaced by study case. Is this graph for “education building” ? The limits of nZEB + and - from minimum is not shown (this will represent limit of Ep) that must be required by RU state. As mentioned before, on Fig 2 is not shown if data are Ep,tot or Ep,nren.

Response 7: The graph does not correspond to any specific building. It has been made as an example. To avoid confusion, the specific values have been removed and "+" and "-" have been included. This is the reason for not including whether it is Ep,tot or Ep,nren, this is a limitation.

Point 8: Chapter 2.2. Limits of Ep,nren and Ep,tot are mentioned, but I can not find the values in discussion and comments. It will be better to use nomenclature from EN 52000-1 (E instead of C). It will be useful that requirements will be shown in Fig. 3.

Response 8: Spanish regulations limit the overall consumption, the abbreviations of these regulations have been kept.

In chapter 3.2, reference is made to these limitations, which are not applicable in the study because only the envelope is being renovated, but which would be applicable when the heating and cooling systems are modified.

Point 9: As suggested in first review, the cash-flow mathematical equation will be helpful for the readers. The example can be, for example, find in paper (DOI: 10.2495/GD170151).

Response 9: Chapter 2.3., We have added the global cost ecuation used (is the ecuation included on Delegated Regulation 244/2012/EU, that is the reason we haven`t written it before)

Point 10: The article would be interesting world-wide. Because of that, the K and qsol must be explained by physical explanation. This was requested in the previous review as well. What is the difference between solar control parameter and solar factor? (last paragraph in 3.2.)

Response 10: we have included in chapter 3.2, both definitions (according to CTE DB-HE).

In this case, solar control and solar factor (in a global meaning) refers the same. We have deleted “solar factor”.

Changes are marked by colours: 

  • Yellow: main changes revision 1
  • Blue: main changes revision 2

Using the "Track Changes" function it is possible to see translation corrections.

Round 3

Reviewer 1 Report

The authors have somewhat improved the paper. The state of the art has been more elaborated, however it is still limited to the few references cited. 

The abstract is still not very clear. 

Overall I would highly recommend English editing by a third party. 

Author Response

Point 1: The authors have somewhat improved the paper. The state of the art has been more elaborated, however it is still limited to the few references cited. 

Response 1: We have added new content.

Point 2: The abstract is still not very clear. 

Response 2: We have removed and changed sentences order. Also we have rewritten some sof them.  

Point 3: Overall I would highly recommend English editing by a third party.

Response 3: We would appreciate if you kindly point out what aspects in the text can be improved in terms of English editing, if relative clauses are too long, if there are spelling mistakes or technical vocabulary was misused, as the other correctors raised no objection in this respect. We need this information to revise the text on a more defined and concrete basis.

Best regards

Reviewer 2 Report

Table 5. you could round 30-years costs to integer

Table 6. check use of "." and "," for decimal separator

Author Response

Point 1: Table 5. you could round 30-years costs to integer

Response 1: We have changed it

Point 2: Table 6. check use of "." and "," for decimal separator

Response 2: We have corrected the mistake